# Comparative Analyses of the Fecal Microbiome of Five Wild Black-Billed Capercaillie (*Tetrao parvirostris*) Flocks

**DOI:** 10.3390/ani13050923

**Published:** 2023-03-03

**Authors:** Xiaodong Gao, Xibao Wang, Xiaoyang Wu, Yongquan Shang, Xuesong Mei, Shengyang Zhou, Qinguo Wei, Guolei Sun, Yuehuan Dong, Weijia Cui, Honghai Zhang

**Affiliations:** College of Life Sciences, Qufu Normal University, Qufu 273165, China

**Keywords:** black-billed capercaillie (*Tetrao parvirostris*), fecal microbiome, function prediction, high-throughput sequencing

## Abstract

**Simple Summary:**

Black-billed capercaillie (*Tetrao parvirostris*) were listed as least concern (LC) class by the International Union for Conservation of Nature (IUCN). However, black-billed capercaillie was an endangered species in China and classified as first-class state protection animal (category I). Only a few studies have focused on the feeding habits of black-billed capercaillie and there is less research content at the molecular level, such as genomics, microbiome, and molecular markers. To improve our understanding of black-billed capercaillie, the fecal microbiome was characterized. This study provides reasonable scientific data for the understanding of black-billed capercaillie and may provide important insights for protecting this endangered species in China.

**Abstract:**

Black-billed capercaillie (*Tetrao parvirostris*) was listed as a first-class state-protected animal because it was endangered in China (Category I). This study is the first to examine the diversity and composition of *T. parvirostris* gut microbiome in the wild. We collected fecal samples from five black-billed capercaillie flock roosting sites (each 20 km apart) in one day. Thirty fecal samples were sequenced with 16S rRNA gene amplicons on the Illumina HiSeq platform. This study is the first to analyze the fecal microbiome composition and diversity of black-billed capercaillie in the wild. At the phylum level, Camplyobacterota, Bacillota, Cyanobacteria, Actinomycetota, and Bacteroidota were the most abundant in the fecal microbiome of black-billed capercaillie. At the genus level, *unidentified Chloroplast*, *Escherichia−Shigella*, *Faecalitalea*, *Bifidobacterium*, and *Halomonas* were the dominant genera. Based on alpha and beta diversity analyses, we found no significant differences in the fecal microbiome between five flocks of black-billed capercaillie. Protein families: genetic information processing; protein families: signaling and cellular processes, carbohydrate metabolism; protein families: metabolism and energy metabolism are the main predicted functions of the black-billed capercaillie gut microbiome through the PICRUSt2 method. This study reveals the composition and structure of the fecal microbiome of the black-billed capercaillie under wild survival conditions, and this study provides scientific data for the comprehensive conservation of the black-billed capercaillie.

## 1. Introduction

Black-billed capercaillie (*Tetrao parvirostris*) belong to the family Tetraonidae (distributed in the coniferous forest zone of eastern Asian Russia). However, black-billed capercaillie were listed as in the class of least concern by the International Union for Conservation of Nature (IUCN), and the species was endangered in China and classified as a first-class state-protected animal (category I). According to historical data, the quantity of black-billed capercaillie in northeast China declined by 35.25% between 1970 and 2018 [1]. Black-billed capercaillie live in flocks through the harsh winter, and their night habitat is dominated by arbor forest (larch, birch, poplar) [2]. Black-billed capercaillie can cope with food shortages using predictable browsing and low energy content foods [3]. During the overwintering period (from October to April of the following year), the bud of Scots pine (*Pinus sylvestris*), Channamu (*Pinus koraiensis*), Larch (*Larix gmelinii*), and Asian white birch (*Betula platyphylla*) are the main constituents of the diets of the black-billed capercaillie [4]. However, there have been few studies on the fecal microbiome of black-billed capercaillie.

The gut microbiome can help the host achieve several important biological functions such as digestion [5], absorption [6], and the maintenance of the gut steady-state [7,8]. Bacteroidetes (phylum), *Cerea bacillus,* and *Pseudomonas aeruginosa* are significantly enriched in the fecal microbiome of geese, which can digest fiber to provide the energy for the host [6,9,10]. The gut microbiome of animals is affected by many factors, including the environment [11,12,13,14], phylogeny [15,16], flocks [17,18], and diet [19,20]. Many studies have focused on the impact of the environment and diet on the gastrointestinal microbiome of different species or the same species. For example, environmental factors affect the alpha and beta diversity of the gut microbial communities of white-crowned sparrows [21]. Noise pollution from urbanization has also significantly changed the gut microbiome and hormone secretion of birds [11]. Based on the cluster analysis and host phylogeny tree, Laviad-Shitrit found that the eco-evolutionary process termed phylosymbiosis may occur between wild waterbird species and their gut microbial community composition [16]. In a broader phylogenetic context, the diet and host phylogeny affect the gut microbial composition of non-passerine birds, with diet being the main influencing factor [15]. In addition, the flight ability can also affect the gut microbiome composition of birds [22]. There have been few studies of the flock factor, and most of them focused on captive environments [17,18].

Therefore, we raise two scientific questions. First, what is the fecal bacteria composition of black-billed capercaillie? Second, is there any difference in fecal bacteria composition between different wild black-billed capercaillie flocks? In this study, we performed 16S rRNA gene V3-V4 region high-throughput sequencing of the fecal microbiome of black-billed capercaillie. We aim ed to explore the fecal bacteria composition of black-billed capercaillie and the composition differences between five wild flocks. We want to provide a reasonable basis for understanding and protecting black-billed capercaillie.

In this article, we first describe the materials and methods used to answer these questions (feces sample collection, bacterial DNA extraction and sequencing, sequence processing, and statistical analyses), then the results of the study (bacterial 16S rRNA gene data, composition of the fecal microbiome, comparison of microbiome for the studied flocks, and the prediction of the gut microbiota function), followed by their discussion and conclusions.

## 2. Materials and Methods

### 2.1. Sample Collection

The fecal samples of black-billed capercaillie were collected on 1 January 2017, in the Jinhe Forestry Bureau of Genhe City, Inner Mongolia. During the sampling period, the temperature was approximately −30 °C, which ensures the freshness of feces. Before sampling, we found the night-roosting habitat of five black-billed capercaillie flocks using transect method. The distance between night-roosting habitats was of 20 km, ensuring that there is no flock duplication. Six fresh feces samples were collected from the several available habitats. We used alcohol (99%) to wipe the scalpel, which was used to cut across the feces placed on the sterile gloves [5]. We only collected the core of the fecal matter to avoid disturbance from other birds and animals [23]. We used sterile gloves to pick up the feces and placed the fecal samples in a sterile tube. This sampling method is from our published article [24]. All samples were stored at −20 °C during the transit to Qufu Normal University, China. We stored all samples at −80 °C until sequencing.

### 2.2. DNA Extraction and High-Throughput Sequencing

The cetyltrimethylammonium bromide (CTAB) method was used to extract the total DNA and the purity and concentration of DNA were then determined using agarose gel electrophoresis (1%) [25].

We used the polymerase chain reaction (PCR) primers (forward primer: CTACGGGNGGCWGCAG; reverse primer: GACTACHVGGGTATCTAATCC) [14,23,26] to amplify the V3–V4 hypervariable regions of bacterial 16S rRNA genes. The PCR amplification volume was 50 μL:5 μL of microbial DNA (5 ng/μL), 25 μL of 2 × Taq PCR Master Mix (0.1 U/μL); 18 μL of double-distilled water (ddH_2_O); 1 μL of forward primer and reverse primer (10 μM). The PCR amplification conditions were as follows: 1 min at 98 °C for pre-denaturation; 25 cycles of 30 s at 95 °C for denaturation; 30 s at 55 °C for annealing and 30 s at 72 °C for elongation, followed by 5 min at 72 °C for the final extension. The PCR products were detected by electrophoresis with 2% agarose gel. We used Qiagen Gel Extraction Kit (Qiagen, Dusseldorf, Germany) to purify the DNA target strip (400–450 bp). We used a TruSeq PCR-Free DNA Sample PreparationKit (Illumina, San Diego, CA, USA) to generate the sequencing libraries and following the manufacturer’s recommendations. We used Qubit 2.0 Fluorometer (ThermoScientific, Waltham, MA, USA) and an Agilent Bioanalyzer 2100 system to evaluate the sequencing library quality. Finally, we used Illumina HiSeq2500 PE250 (San Diego, CA, USA) to sequence the sequencing library.

### 2.3. Sequence Processing and Statistical Analyses

We used Fast Length Adjustment of Short reads (FLASH software; V 1.2.11) to cut the barcode and primer sequences and combined the pair-ended reads. We referred to the tags quality control process of Quantitative Insights into Microbial Ecology (QIIME) pipeline (Qiime; V1.7.0) pipeline and performed the following operations: tags interception and tags length filtering. Based on UCHIME algorithm [20], the Parallel-Meta Suite (PMS; V 3.7) [27] was used to eliminate chimeric sequences. Based on the sequence similarity at 97%, we performed PMS to cluster the sequences into operational taxonomic units (OTUs). We used vsearch [28] and SILVA ribosomal RNA databases (v 138.1) built into PMS (V 3.7) to annotate each sample, and we then obtained the set of operational taxonomic units (OTUs) relative abundance table and each taxon relative abundance tables for all data. Based on the OTU level, the rarefaction curve, species accumulation boxplot, non-metric multidimensional scaling analysis (NMDS), principal coordinate analysis (PCoA), and Tukey test of beta diversity were plotted using the ggplot2 package in R software (v 4.2.1). The Kruskal–Wallis test of alpha diversities (richness, chao1, and Good’s coverage indices) among five wild flocks was calculated using the vegan package of R software (v 4.2.1). Rarefaction curve and Good’s coverage index can be used to evaluate whether the sequencing results and depth content meet the requirements of subsequent experiments. The species accumulation boxplot can be used to evaluate whether the sequencing samples are sufficient for analysis. We used PCoA, NMDS, and the Tukey test of beta diversity to verify whether the bacterial compositions of five flocks were different. We used the Phylogenetic Investigation of Communities by Reconstruction of Unobserved States (PICRUSt2) software [29] to predict the gut microbiome functions of black-billed capercaillie. In all figures, we represent the samples of black-billed capercaillie by “HZ” and the five flocks by “F” (F1, F2, F3, F4, and F5).

## 3. Results

### 3.1. 16S rRNA Gene Data

We obtained 2,190,160 effective tags from 30 black-billed capercaillie samples, with an average of 73,005 (standard deviation, 9509) sequences per sample. The 30 rarefaction curves (Figure 1) and species accumulation boxplot (Figure 2) became gradually flattened, indicating that the sequencing depth and the number of samples were reasonable for the subsequent experimental analyses. The goods coverage index (Figure 3) of all data was higher than 99.6%, suggesting that all sequencing data effectively represent the fecal microbiome of black-billed capercaillie.

### 3.2. Results for Fecal Microbiome

At the phylum level, we used bacteria with >1% abundance in each sample to generate a relative abundance column cumulative plot. Camplyobacterota (33.44%) were the most abundant in the gut microbiome of black-billed capercaillie, and the other fecal dominant bacteria of gut microbiome were Bacillota (25.51%), Cyanobacteria (24.12%), Actinomycetota (14.92%), Bacteroidota (1.46%), and Acidobacteriota (0.26%) (Figure 4A).

We listed OTUs with an average abundance greater than 1% in Table 1. Among them, six OTUs (OTU_3504_102, OTU_3327_4, OTU_3616_27, OTU_26443_5, OTU_3400_101, and OTU_3616_340) belong to Camplyobacterota, three (OTU_11392_1, OTU_2185_14, and OTU_2184_5) belong to Bacillota, two (OTU_25228_508 and OTU_25228_2257) to Cyanobacteria, and two OTUs (OTU_436_66 and OTU_13901) belong to Actinomycetota.

At the genus level, we used the abundance of the top 10 genera to generate a relative abundance column cumulative plot. The *Faecalitalea* (12.60%, belong to Bacillota) were the most abundant in the gut microbiome of black-billed capercaillie. *Escherichia−Shigella* (10.67%, belong to Camplyobacterota), *Halomonas* (5.78%, belong to Camplyobacterota), *Bifidobacterium* (5.65%, belong to Actinomycetota), and *Anaerobiospirillum* (3.38%, belong to Camplyobacterota) were the other major genera in the gut microbiome of black-billed capercaillie (Figure 4B). The relative abundance of the top 10 bacteria genera occupied more than 38% in the fecal microbiome relative abundance of the black-billed capercaillie.

### 3.3. Analysis of Differences between the Flocks

The richness and chao1 indices can show the alpha diversity of the fecal microbiome, and the Kruskal–Wallis test examined the difference in the alpha diversity of the fecal microbiome of five black-billed capercaillie flocks. The boxplot showed that the indexes of richness (*p* = 0.42) and chao1 (*p* = 0.6) were not significantly different between the five wild flocks (Figure 3). Based on the unweighted and weighted unifrac distance, the beta diversity boxplot showed that the fecal microbiome composition of five wild flocks were not significantly different (Figure 5; Tukey test; *p* > 0.05). The NMDS (Figure 6A) and PCoA (Figure 6B) also indicated that five wild flocks of black-billed capercaillie were grouped together. The results suggest that the fecal microbiome diversity was not significantly different between the five wild flocks.

### 3.4. Prediction of Gut Microbiome Function

Based on the functional prediction analysis, we predicted the 53 KEGG Level 2 categories. All samples displayed high abundances in KEGG Level2 categories of the following protein families: genetic information processing (15.97%); protein families: signaling and cellular processes (5.58%), carbohydrate metabolism (8.75%), amino acid metabolism (6.64%); protein families: metabolism (6.57%), energy metabolism (4.55%), metabolism of cofactors and vitamins (4.11%), membrane transport (1.35%), and translation (2.87%) (Appendix A).

## 4. Discussion

In this study, we first characterized the gut microbiome and potential functions of black-billed capercaillie using the 16S rRNA gene. The relative abundance of Camplyobacterota, Bacillota, Cyanobacteria, Actinomycetota, and Bacteroidota were the major phyla in gut microbiome of black-billed capercaillie. These results were basically consistent with the previous characterizations of birds’ gut microbiome, such as that of Oriental stork (*Ciconia boyciana*) [12], *Zonotrichia leucophrys nuttalli* [11], and crested ibis (*Nipponia nippon*) [30].

There was no significant difference in the alpha and beta diversity of the gut microbiome between the five wild flocks. Gut bacterial diversity was generally associated with the food diversity. There may not have been any appreciable variations in the gut microbiota makeup of the five wild flocks due to the same forest environment, analogous vegetation (similar diet), and severely cold temperatures.

Camplyobacterota contains the most OTUs with an average abundance greater than 1% and was the most abundant phylum enriched in the gut microbiome of black-billed capercaillie. Camplyobacterota was a highly functionally diverse group of bacteria and related to the functional variability of the gut [31], and the most abundant phylum in fecal microbiome of many birds [15,22,31]. Camplyobacterota can maximize the function of the microbiome while reducing the diversity and mass [12]. This may be beneficial for the black-billed capercaillie to adapt to the buds of various trees (Scots pine, Channamu, Larch, and Asian white birch). The OTU_436_66 and OTU_24538_7 belong to Actinomycetota, which is associated with carbohydrate metabolism [32]. Bacillota contains three highly abundant OTUs: OTU_11392_1, OTU_2185_14, and OTU_2184_5. The food composition of black-billed capercaillie contains fiber and cellulose, and Bacillota contains a lot of fiber-decomposing bacteria and can decompose the fiber and cellulose to provide energy for the host [33,34].

The highly abundant OTU_25228_508 and OTU_25228_2257 belong to Cyanobacte-ria, which are enriched in the gut microbiomes of passerines [35]. Based on research on the gut microbiome of the American white ibis (*Eudocimus albus*), it was found that the relative abundance of Cyanobacteria is negatively correlated with urban land cover [36]. n our study, Cyanobacteria was the third-most dominant phylum in the fecal microbiome of the Black-billed capercaillie. We speculate that China’s policy of returning farmland to forests has effectively increased its habitat. This policy may prove to be beneficial to in-crease the wild Black-billed capercaillie population.

OTU_11392_1 was annotated as *Faecalitalea*, which is isolated from the feces of chickens, and can produce butyric acid, lactic acid, and formic acid as the main metabolic end products [37]. These acids are important energy substances for the host and maintain host intestinal homeostasis [38,39]. In our study, the OTU_436_66 in the gut microbiome of black-billed capercaillie was annotated as *Bifidobacterium*. Complex carbohydrates are the substrate for *Bifidobacterium*, which are the beneficial bacteria in the host’s gut [40,41]. These two highly abundant bacteria contribute to the energy acquisition and maintenance of gut homeostasis in black-billed capercaillie.

Two OTUs were annotated as *Halomonas*, with a sum of abundances of 3.76%. *Halo-monas* can balance the cell’s osmotic pressure and protect the structures of enzymes, DNA, and the cytomembrane that may improve cell motility, amino acid and carbohydrate metabolism, and the environmental adaptation of the black-billed capercaillie [42]. In our study, OTU_3504_102 was annotated as *Escherichia−Shigella*, which comprises the dominant bacteria in the gastrointestinal tract microbiomes of newly hatched chicks [43], Grey catbird (*Dumetella carolinensis*), and Swainson’s thrush (*Catharus ustulatus*) [44]. In general, *Escherichia−Shigella* is considered pathogenic bacteria in the animal gut microbiome. For example, *Escherichia−Shigella* is negatively correlated with the gut microbi-ome alpha diversity and weights of female steppe buzzards [45]. *Ralstonia* was present as a highly abundant OTU (OTU_26443_5), which uses volatile fatty acids as substrates [46]. *Ralstonia* acts as a pathogen or opportunistic pathogen in the host [47]. However, we did not conduct metagenomic sequencing analysis and Black-billed capercaillie health assessments. Therefore, we could not determine whether *Escherichia−Shigella* and *Ralstonia* are pathogenic bacteria in the gut microbiomes of the Black-billed capercaillie.

The predictions of fecal microbiome functions for the Black-billed capercaillie were performed using the PICRUSt2 method [29]. Carbohydrate metabolism, protein families: metabolism, and energy metabolism were the functions found to be highly abundant based on functional prediction analysis. In an environment with a lack of food resources, the predicted fecal microbiome functions were found to be almost consistent among the five wild black-billed grouse flocks. These predicted functions could allow the black-billed grouse to obtain more energy for life activities in the buds of Scots pines (*P. sylvestris*), Channamu (*P. koraiensis*), larch (*L. gmelinii*), and Asian white birch (*B. platyphylla*). These findings will help us to better understand how the fecal microbiome and metabolism interact. However, as this was not a real metagenomic sequencing analysis, metagenomics and transcriptomics should be analyzed in the future.

## 5. Conclusions

Through 16S rRNA gene sequencing analysis, we characterized the gut microbiome composition of the Black-billed capercaillie and analyzed the differences between flocks. Camplyobacterota, Bacillota, Cyanobacteria, Actinomycetota, and Bacteroidota were the major phyla in the fecal microbiome of the Black-billed capercaillie, and we found no no-table differences in the fecal microbiome compositions, diversity, and predictive functions among different wild flocks. Genetic information processing, signaling and cellular pro-cesses, carbohydrate metabolism, amino acid metabolism, metabolism, energy metabo-lism, and metabolism of cofactors and vitamins were the main functions in the gut micro-biomes of different wild flocks. We hypothesize that the reason for this is the extreme cold conditions and similar tree compositions (same diet) between habitats. These results will help researchers further understand the wild Black-billed capercaillie and contribute to the comprehensive conservation of this endangered species.

## Figures and Tables

**Figure 1 animals-13-00923-f001:**
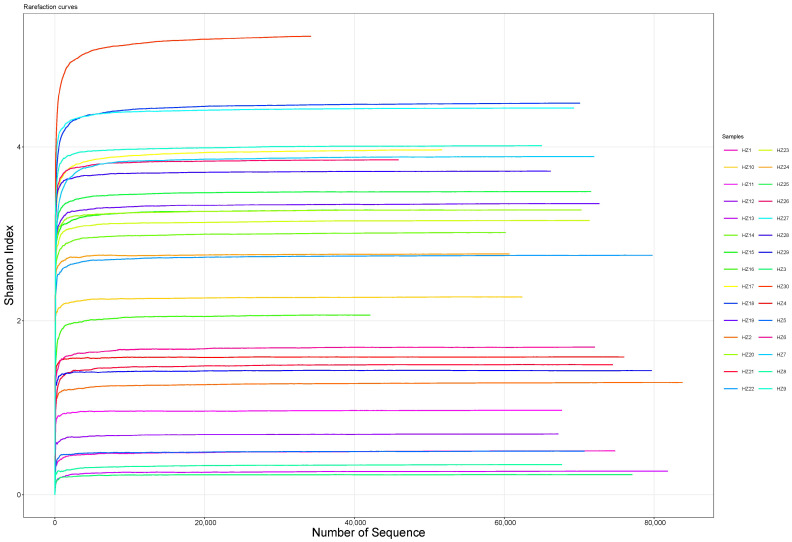
Rarefaction curve plot of 30 black-billed capercaillie samples. In the rarefaction curves, the abscissa is the number of sequencing samples randomly chosen from the sample, and the ordinate is the number of OTUs. Each line represents the gut microbiome of a black-billed capercaillie sample.

**Figure 2 animals-13-00923-f002:**
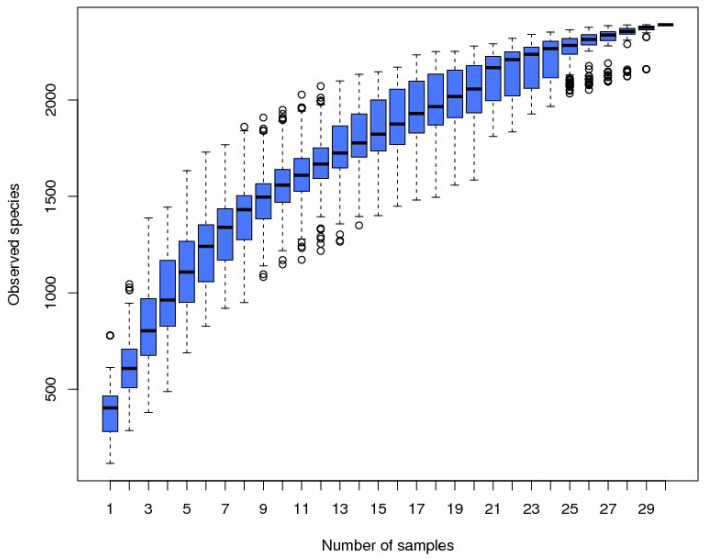
Species accumulation boxplot. The abscissa is the number of black-billed capercaillie samples, and the ordinate is the number of observed species. The species accumulation boxplot reaches a plateau indicating that the number of OTUs in the community does not increase with an expanding number of black-billed capercaillie samples.

**Figure 3 animals-13-00923-f003:**
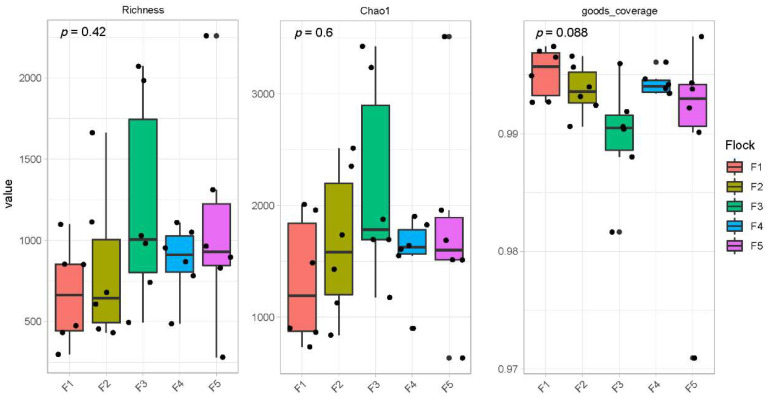
Kruskal–Wallis test of fecal microbiome alpha diversity between five wild black-billed capercaillie flocks. The richness and chao1 indices indicate the number of species within the community (higher values indicate higher diversity and richness of the community). The goods coverage index evaluates whether the sequencing results represent the true situation of the fecal bacteria community (the value is close to 1, and the sequencing depth basically covers all species in the sample). The abscissa is the group size and the ordinate is the alpha diversity value.

**Figure 4 animals-13-00923-f004:**
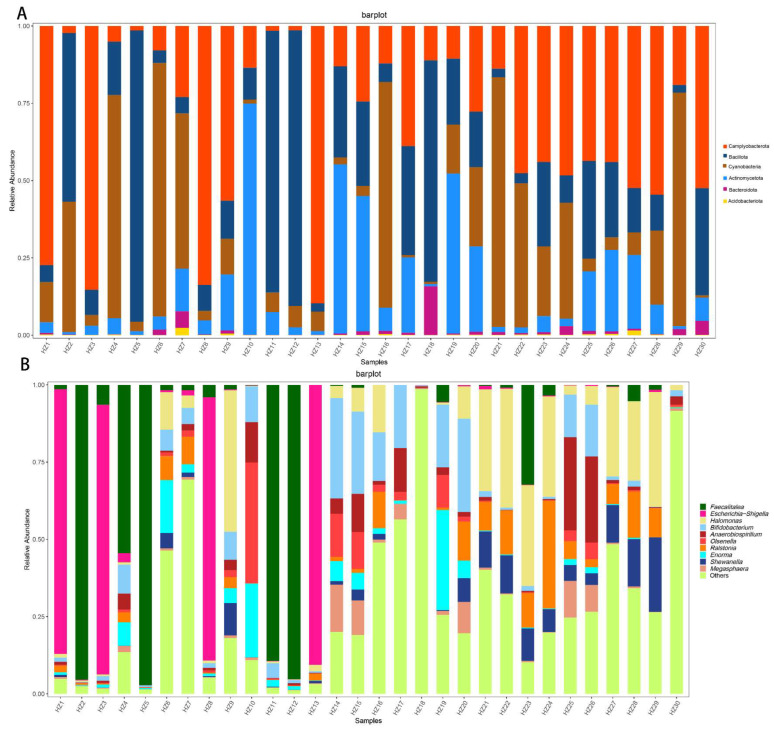
Gut microbiome composition between five wild black-billed capercaillie flocks at the phylum (**A**) and genus (**B**) levels. The abscissa represents the black-billed capercaillie samples, and the ordinate represents the relative abundance of fecal bacteria.

**Figure 5 animals-13-00923-f005:**
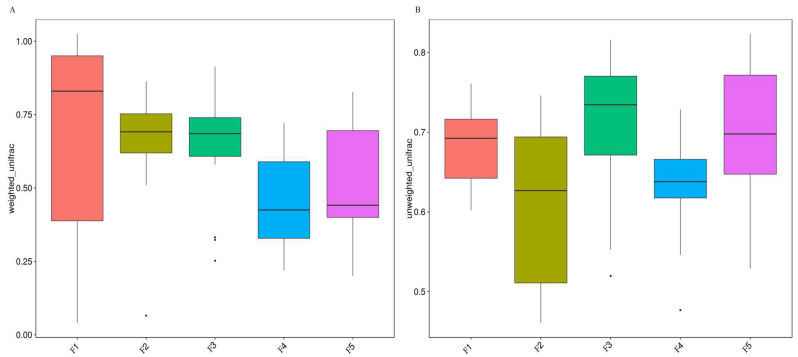
Comparisons for beta-diversity between five wild flocks based on the weighted (**A**) and unweighted (**B**) unifrac distance. The abscissa is the group size, and the ordinate is the unweighted or weighted unifrac distance.

**Figure 6 animals-13-00923-f006:**
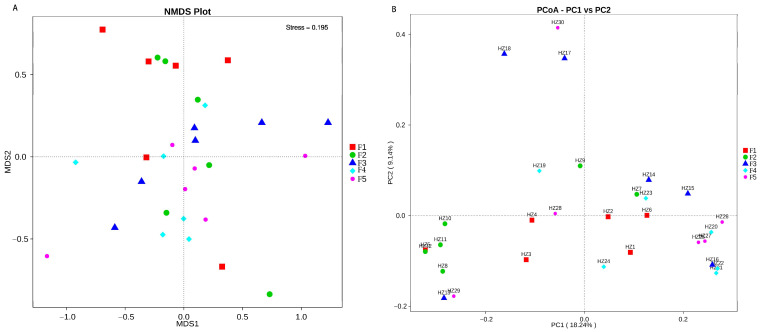
Non-metric multidimensional scaling analysis (NMDS) (PCA; (**A**)) and the principal coordinate analysis (PCoA; (**B**)) of the fecal microbiome composition. Each point represents the fecal microbiome of a black-billed capercaillie sample.

**Table 1 animals-13-00923-t001:** The OTUs with an average abundance greater than 1% and their classification level.

OTU	Abundance (%)	Phylum	Class	Order	Family	Genera
OTU_25228_508	13.80	Cyanobacteria	Oxyphotobacteria	Unidentified	Unidentified	Unidentified
OTU_11392_1	12.56	Bacillota	Erysipelotrichia	Erysipelotrichales	Erysipelotrichaceae	*Faecalitalea*
OTU_3504_102	10.58	Camplyobacterota	Gammaproteobacteria	Enterobacteriales	Enterobacteriaceae	*Escherichia-Shigella*
OTU_25228_2257	9.94	Cyanobacteria	Oxyphotobacteria	unidentified	unidentified	unidentified
OTU_436_66	5.61	Actinomycetota	Actinobacteria	Bifidobacteriales	Bifidobacteriaceae	*Bifidobacterium*
OTU_3327_4	3.37	Camplyobacterota	Gammaproteobacteria	Aeromonadales	Succinivibrionaceae	*Anaerobiospirillum*
OTU_3616_27	2.72	Camplyobacterota	Gammaproteobacteria	Oceanospirillales	Halomonadaceae	*Halomonas*
OTU_26443_5	2.69	Camplyobacterota	Gammaproteobacteria	Betaproteobacteriales	Burkholderiaceae	*Ralstonia*
OTU_13901	2.62	Actinomycetota	Coriobacteriia	Coriobacteriales	Coriobacteriaceae	*Enorma*
OTU_3400_101	2.24	Camplyobacterota	Gammaproteobacteria	Alteromonadales	Shewanellaceae	*Shewanella*
OTU_24538_7	2.13	Actinomycetota	Coriobacteriia	Coriobacteriales	Atopobiaceae	*Olsenell*
OTU_2185_14	1.92	Bacillota	Negativicutes	Selenomonadales	Veillonellaceae	*Megasphaera*
OTU_2184_5	1.27	Bacillota	Negativicutes	Selenomonadales	Veillonellaceae	*Megasphaera*
OTU_3616_340	1.04	Camplyobacterota	Gammaproteobacteria	Oceanospirillales	Halomonadaceae	*Halomonas*

## Data Availability

The raw data presented from this study can be found in the Genome Sequence Archive in BIG Data Center (accession number, CRA008095; https://ngdc.cncb.ac.cn/, accessed on 7 September 2022) and the Sequence Read Archive in National Center for Biotechnology Information (accession number, PRJNA876315; https://www.ncbi.nlm.nih.gov, accessed on 2 September 2022), respectively.

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
