# Peer review of "Comparative Analyses of the Fecal Microbiome of Five Wild Black-Billed Capercaillie (Tetrao parvirostris) Flocks"

_animals, 2023, doi:10.3390/ani13050923_

Round 1

Reviewer 1 Report

Dear authors,

please find the comments in the attached pdf file.

Author Response

We have carefully studied your valuable comments and have tried our best to revise the manuscript accordingly. We modified the preparation of this manuscript and have used the services of Editage (https://accounts.editage.com/) for linguistic assistance. The point to point responds are listed as following:

Line 1: As I understand it, not the gut microbiome but rather the fecal microbiome was

characterized. Please correct this statement.

Response: Thank you for your valuable advice. We have changed the title in Line 1.

Line 16: “These is less research content on molecular level”. Please clarify this statement. What molecular level?

Response: Thank you for your careful work and valuable advice. We added the molecule techniques in Line 16.

Line 17: In this study, the fecal microbiome was characterized.

Response: Thank you for your careful work and valuable advice. We modified the sentence in Line 17.

Line 18: Conservation of T. parvirostris? I cannot understand this statement please elaborate.

Response: I am sorry for the confusion caused by my unclear description. My intention was to provide insights into the protection of T. parvirostris. We modified the sentence in Line 18.

Line 21: In this study, the fecal microbiome was characterized.

Response: Thank you very much for your valuable advice. We changed the “gut” to “fecal” and moved the sentence in the Line 26.

Line 27: Should be based on alpha and beta diversity analyses.

Response: Thank you very much for your valuable advice. I am sorry for the confusion caused by my unclear description. We modified the sentence in the Line 30. 

Line 30: Please be clear about the capabilities of Picrust2 to only estimate the function.

Response: Thank you for your careful work. We modified the sentence in the Line 33.

Introduction: The introduction provides extensive insight into various projects regarding mammals that the authors themselves conducted. In general, this is fine, however, comparing mammals with birds regarding their digestion does not make sense as the bird’s digestive system is vastly different from that of in the examples supplied mammals. Furthermore, as the bird digests mainly nuts and seeds, it does make even less sense to compare it with meat or grass eating mammals such as Ref. 1 the fox, Ref. 5 the horse, Ref. 6 deer, Ref. 7 ungulates, Ref. 8. Marine carnivores, Ref. 9 gorillas, Ref. 10 humans, Ref. 11 panda, Ref. 12 monkeys. I would rather have read about the microbiome in goose or other bird species. There is extensive literature regarding the microbiome of chickens and goose, I have added some examples below, however, there are many more publications and please feel not obliged to choose these.

Response: Thank you for pointing out the error. We deleted some references of mammals and added references of avian according to your recommendation. We have made a lot of changes to this paragraph.

Line 41: Please clarify what is meant by niche here. Line 42: Please clarify what is meant by niche here.

Response: I am sorry for my unclear description. This “niche” is feeding niche. However, I rewrote this paragraph and changed the “niche” to the “environment”.

Line 58 to 60: I find this information very helpful for the reader and would rather base the introduction on specific knowledge about the bird’s behavior and diet and then introduce ideas from other birds that might be helpful to the reader.

Response: Thank you for your valuable advice. We added more basic research about T. parvirostris and gut microbiome of other birds.

Line 61: I don’t understand question 1. Are you asking if there is a bacterial community in the microbiome or rather what is the bacterial community?

Response: I am sorry for the confusion caused by my unclear description. We modified the sentence in line 86.

Line 64: Please be very clear on what was analyzed. I understood from the introduction that fecal samples have been used so this is not about the gut microbiome but about the fecal microbiome.

Response: I am sorry for the confusion caused by my unclear description. We changed the gut to fecal.

Line 69: Could the authors please provide information on how the samples were taken. Did you use specific tubes or stabilizer? How did you omit contamination p.e. by birds or other animals? Have all samples been collected on the same day?

Response: Thank you for your valuable advice. We used alcohol (99%) to wipe the scalpel, which is used to cut across the feces placed on the sterile gloves. We only collected the core of fecal to avoid disturbance from other birds and animals. We used sterile gloves to pick up the feces and placed the fecal samples in a sterile tube. We added the information in Line 94-106.

All samples have been collected on the same day. We have three teams, so we can collect all the samples in one day.

Line 99: Were further quality ensuring procedures used? P.e. trimming reads at the end and

beginning, barcode mismatches, abundance filters,…

Response: Thank you for your careful work. We neglected to describe such methods in the manuscript. We have added the information in manuscript.

Line 107: Here, T. parvirostris is twice written non italic. Could the authors please provide a statistic section to the Materials and Methods part.

Response: Thank you for your careful work. We modified the writing errors.

Line 111: Could the authors please provide the standard deviation for the average sequences per sample?

Response: Thank you for your valuable advice. We added the standard deviation for the average sequences in the manuscript.

Figure 1: Could the authors please increase the text size in this figure. I cannot decipher it.

Response: I am sorry for the low quality figure. We have modified figure 1.

As far as I can see it, there is no explanation or naming of Figures 2 and 3 in the text. Please provide this.

Response: I am sorry for no explanation or naming of Figures 2 and 3. We added the explanation in title of figure 2 and 3.

Figure 3: The text above the boxplot is very hard to read (i.e. Richness, Chao1, ACE, etc).

Further, p values and p should be italicized.

Response: I am sorry for the low quality figure. We modified the “Richness, Chao1” and p values in figure 3.

Lines 127 – 132: The phyla have recently been renamed and published. Please use the names

according to the ICNP (Oren A, Garrity GM. Notification list. Notification that new

names and new combinations have appeared in volume 71, part 10 of the IJSEM.

Int J Syst Evol Microbiol 2022; 72:5165.

Response: Thank you for your valuable advice. We have used the names according to the ICNP and modified the figure and sentence.

Lines 133 – 137: Genra are italicized commonly, please change accordingly.

Response: Thank you for your careful work. We modified the writing errors.

Line 137: These genera were not other major bacteria but other major genera

Response: Thank you for your careful work. We modified the writing errors.

In general, T. parvirostris is non-italicized multiple times, please change.

Response: Thank you for your careful work. We modified the writing errors.

Figure 4. Please enlarge this figure as I cannot read the legend as it is

Response: I am sorry for the low quality figure. We enlarged this figure in the manuscript.

Line 143: Please explain to the reader what these diversity parameters indicate and why you used them. Especially the “goods_coverage” is not known to me.

Response: Thank you for your valuable advice. We added the diversity parameters indicate and reason for selection in figure 3 and line 179-180, and deleted ACE index (Because it overlaps with the indicate of the chao1 index).

Figure 5: Please enlarge the text in this figure.

Response: I am sorry for the low quality figure. We enlarged this figure in the manuscript.

Figure 7: Footer: Is the explanation wrong? I cannot read the text in this figure, however, this should be about the function? Could the authors give some exact enzymes they found.

Response: I am sorry for the wrong title of figure 7. I have modified the title. We would have liked to look for some exact enzymes, however it was difficult to go for them as they were functionally predicted. We will continue to explore the functions and enzymes of the fecal microbiome based on Metagenomics in the future. We have placed figure 7 in the Supplementary Material so that it can be enlarged and the text on the figure can be clearly seen.

Line 167 16S rRNA

Response: Thank you for your careful work. We modified the writing errors.

Line 167/8: Why would the authors sort the phyla in this manner, when in the results it was stated that Pseudomonadota (previously Proteobacteria) dominated the microbiome? Please provide the correct sortation according to the relative abundance.

Response: Thank you for your careful work. We have adjusted the order.

Line 174: Why do the authors compare birds to fish or insects? Please omit from such comparisons and stay with birds.

Response: Thank you very much. We added the references of bird in the line 223.

Line 177-185: Here, the authors guess about the functions of phyla regarding the microbiome. Phyla are incredibly diverse and should not be used for such comparisons. It is sometimes even hard to compare on family or genus level. I would advise the authors to look on the OTU level and check the most abundant OTUs and explain matters according to the most similar species to an OTU.

Response: Thank you very much for your guidance. We added the OTU with average abundance table in supplementary material and explained matters according to the most similar species to an OTU.

Line 186 – 196: See Line 177-185. Please only guess functions based on OTUs.

Response: Thank you very much for your guidance. We added the OTU with average abundance table in supplementary material and explained matters according to the most similar species to an OTU.

Line 192: Please be very careful with the statement, that a certain genus has been found in the embryo. There is increasing evidence, that such environments are sterile and sequencing results most probably have been contaminations. Line 193: Halomonas … are related to genetic? I do not understand this statement.

Response: Thank you very much. We have read the literature and you are right. We deleted relevant discussions to avoid disputes.

Line 197: Please be very careful here. Picrust2 does not provide the exact functions but

rather estimates the functions of the microbiome based on the phylogenetic composition of the samples. Please be very clear about this matter.

Response: Thank you very much. We added "prediction" and "functional prediction" in the manuscript. We will try to avoid the mistake of unclear description in the future.

High abundance of Proteobacteria and an unclassified genus of Choloroplast were observed, please discuss why this is the case. This is quite unusual for bird fecal microbiome.

Response: Thank you very much for your guidance. We rechecked the analysis steps and found that the PMS database download was incomplete. So, we re-downloaded the database and confirmed that the database was complete. After another analysis, the phylum Camplyobacterota was still the most abundant phylum, while the genus with the highest taxonomic level was Faecalitalea. The phylum Camplyobacterota is the most abundant phylum in the fecal of many birds, such as Gruiformes, Anseriformes, and Ciconiiformes in the research “Covariation of the Fecal Microbiome with Diet in Nonpasserine Birds”. We have discussed and cited in the article.

We appreciate your review and comments on the article. It enhances the quality of our articles. Thank you very much for giving us the opportunity to make this revision. We hope that this revision will meet your expectations

Reviewer 2 Report

Dear Sir

I have enclosed my comments for kind perusal and correction

kind regards

akt

Author Response

We have carefully studied your valuable comments and have tried our best to revise the manuscript accordingly. We modified the preparation of this manuscript and have used the services of Editage (https://accounts.editage.com/) for linguistic assistance. The point to point responds are listed as following:

Reviewer #2:

Line 14 for conserva-tion of nature (IUCN) for conservation of nature (IUCN).

Response: Thank you for your review. We modified the sentence in the simple summary.

Abstract: line 20-22, Tetrao parvirostris was endangered in China and under first-class state protection animal (category I). This study is the first to analyze the gut microbiome composition and diversity of T. parvirostris in the wild.

Response: Thank you for your careful work and valuable advice. We modified the sentence in the abstract.

Line 39: Firmicutes is significant enriched in rumen.

Response: Thank you very much. According to your comment, we have changed the “significant” to “significantly” in line 63.

Line 50: However, there are few studies on flock factor. Especially different flocks of birds 50 in the same area

Response: Thank you very much. According to your comment, we have modified the sentence in line 85.

A few lines of utility of the study may be clearly indicated with future direction.

Response: Thank you very much for your guidance. We added the direction in line 94.

Line 79-81: Thank you very much for your guidance. Total DNA was extracted using cetyltrimethylammonium bromide (CTAB) method. Then the purity and concentration of DNA were detected by agarose gel electrophoresis (1%).

Response: Thank you for your careful work and valuable advice. We modified the sentence in line 112-114.

Line 127-132: At the phylum level, we used bacteria with abundance top 10 phyla to generate a relative abundance column cumulative plot. Proteobacteria (35.87%) were the most abundance in the gut microbiome of T. parvirostris. Other dominated bacteria of gut microbiome were Firmicutes (23.60 %), Cyanobacteria (20.29 %), Actinobacteria (16.25 %), and Bacteroidetes (2.56 %) were the other major bacteria in gut microbiome of T. parvirostris (Fig. 4A).

Response: Thank you for your careful work and valuable advice. We modified the sentence in line 173-180.

Line 127-132: The boxplot showed that the index of richness (p = 0.42), chao1 (p = 0.6), ACE (p = 0.43), and goods_coverage (p = 0.088) weren’t significantly differences between five wild flocks (Fig. 2). The results suggesting that gut microbiome diversity and richness weren’t significantly difference between 5 flocks. Based on the unweighted and weighted unifrac distance, the beta boxplot showed that the gut microbiome composition of five wild flocks weren’t significantly difference (Fig. 5; Tukey test).

Response: Thank you for your careful work and valuable advice. We modified the sentence in line 190-198

Line 172-176: There was no significant difference in the alpha and beta diversity of gut microbiome between the five wild flocks. Gut bacterial diversity generally associate with the food diversity [27,28]. The same forest environment with the same trees (same diet) and extremely cold weather may have resulted in no significant differences in the gut microbiome composition of the 5 flocks.

Response: Thank you for your careful work and valuable advice. We modified the sentence in line 231-235.

Line 182: Firmicutes are the one of the most important.

Response: Thank you for your careful work and valuable advice. Based on suggestions from other reviewers, we revised and deleted this sentence.

Line 200-203: These data contribute to the understanding of the relationship between gut microbiome and metabolism. However this is not a true metagenomic sequencing analysis. Therefore, we need to continue our research in the future by metagenomics and transcriptomics.

Response: Thank you for your careful work and valuable advice. We modified the sentence in line 270-272.

Conclusion Not informative and may be refined.

Response: Thank you for your careful work and valuable advice. We added some information in conclusion.

Line 211-212: We hope to contribute to the rescue and ex situ conservation of the T. parvirostris through this study.

Response: Thank you very much. We have deleted the sentence.

Line 214-215: For research articles with several authors, a short paragraph specifying their individual contributions must be provided. The following statements should be used.

Response: Thank you very much. We have deleted the sentence. We are very grateful for your valuable comments, which have improved the quality of our articles. Again, I would like to express my sincere thanks.

Reviewer 3 Report

Line 15-16 that sentence can be written better. 

Line 56 October is misspelled

3.2 Results for Gut Microbiome 

Why only present data from phyla and genus from the 16s metagenomic studies?  

Was Halomonas found in your study?  

How did you come to the conclusion that the difference between the clusters is due to extreme cold conditions? 

Author Response

We have carefully studied your valuable comments and have tried our best to revise the manuscript accordingly. We modified the preparation of this manuscript and have used the services of Editage (https://accounts.editage.com/) for linguistic assistance. The point to point responds are listed as following:

Line 15-16 that sentence can be written better.

Response: I'm sorry for bad sentence. We have modified the sentence.

Line 56 October is misspelled

Response: I'm sorry for spelling mistakes. We have modified the word.

3.2 Results for Gut Microbiome

Response: Thank you very much. We have modified the title.

Why only present data from phyla and genus from the 16s metagenomic studies?  

Response: Thank you very much. Demonstration of phylum taxonomic levels is an overall characterization of the fecal microbiome of T. parvirostris. Because 16S rRNA sequencing is difficult to annotate to the species taxonomic level. So, we show the genus classification level.

Was Halomonas found in your study?

Response: Yes, in our study, OTU_3616_27 and OTU_3616_340 were annotated as Halomonas.

How did you come to the conclusion that the difference between the clusters is due to extreme cold conditions? 

Response: I am sorry for the confusion caused by my unclear description. We found no significant differences in the fecal microbiome composition, diversity, and predictive functions among the different flocks. We hypothesize that reason is due to extreme cold conditions and similar tree composition (same diet) between habitats. Because diets and temperature are two important factors that determine the fecal microbiome of wild flocks. We are very grateful for your valuable comments, which have improved the quality of our articles. Again, I would like to express my sincere thanks.